# Study on the Influence of Mounting Dimensions of PV Array on Module Temperature in Open-Joint Photovoltaic Ventilated Double-Skin Façades

**Wenjie Zhang \*, Tongdan Gong, Shengbing Ma, Jianwei Zhou and Yingbo Zhao**

School of Energy and Power Engineering, Nanjing University of Science and Technology, Nanjing 210094, China; gtd1998@126.com (T.G.); 13688843389@163.com (S.M.); zhjwhvac@njust.edu.cn (J.Z.); zhaoyingbo9611@126.com (Y.Z.)
* Correspondence: zhangwenjie001@139.com

**Abstract:** In building integrated photovoltaics (PV), it is important to solve the heat dissipation problem of PV modules. In this paper, the computational fluid dynamics (CFD) method is used to simulate the flow field around the open-joint photovoltaic ventilated double-skin façades (OJ-PV-DSF) to study the influence of the mounting dimensions (MD) of a PV array on the module temperature. The typical summer afternoon meteorological parameters, such as the total radiation (715.4 W/m$^2$), the outdoor temperature (33.1 °C), and the wind speed (2.0 m/s), etc., are taken as input parameters. With the DO (discrete ordinates) model and the RNG (renormalization-group) $k - \varepsilon$ model, a steady state calculation is carried out to simulate the flow of air in and around the cavity under the coupling of hot pressure and wind pressure, thereby obtaining the temperature distribution of the PV array and the wall. In addition, the simulation results are compared with the onsite experimental data and thermal imaging to verify the accuracy of the CFD model. Then three MD of the open joints are discussed. The results show that when the *a* value (represents the distance between PV modules and wall) changes from 0.05 to 0.15, the temperature drop of the PV module is the most obvious, reaching 2.0 K. When the *b* value (representing the distance between two adjacent PV modules in the vertical direction) changes from 0 to 0.1, the temperature drop of the PV module is most obvious, reaching 1 K. When the *c* value (represents the distance between two adjacent PV modules in the horizontal direction) changes from 0 to 0.1, the temperature of the PV module is lowered by 0.8 K. Thus, *a* = 0.1–0.15, *b* = 0.1 and *c* = 0.1 are recommended for engineering applications to effectively reduce the module temperature.

**Keywords:** photovoltaic double skin façades (PV-DSF); air layer; PV array arrangement; CFD; module temperature

## 1. Introduction

In China, the proportion of building energy consumption to total social energy consumption is close to 30% [1], while heating and air-conditioning system energy consumption accounts for 40–50% of building energy consumption [2]; and the demand for indoor thermal comfort is also becoming higher with the improvement of people's living standards. Therefore, the energy consumption of building heating and air-conditioning systems will gradually increase. Reasonable reduction of energy consumption in heating and air-conditioning systems is one of the important ways to achieve energy efficiency in buildings. However, in modern architecture, in pursuit of a beautiful appearance, designers often choose glass curtain walls as the building envelope, which also leads to a sharp increase in the energy consumption of the heating and air-conditioning system. In recent years, with the maturity of photovoltaic (PV, see Appendix A, Table A1) technology, the continuous innovation of PV cell types and the continuous improvement of component efficiency, the application of PV technology in buildings is continuously enriched; the integration with

the building curtain wall is an ideal combination; the PV façade [3] not only looks good, but also has many additional benefits. It not only reduces the energy consumption of building heating and air conditioning in a passive way, but also actively produce a part of the electric energy, thus reducing the building's dependence on external conventional energy [4].

However, there are still many problems to be solved in the application process of a PV façade. For example, under solar radiation, PV modules generate waste heat while generating electricity [5], and are susceptible to their own temperature coefficients [6]. When integrated with buildings, they may cause poor heat dissipation, which will increase the operating temperature of the PV modules, thereby reducing their electrical performance [7,8]. On the other hand, as a building envelope, the thermal performance of the PV façade will directly affect the heat gain of the building interior, which in turn affects the energy consumption of air conditioners, lighting and other equipment [9]. Therefore, it is particularly important to take effective measures to increase the heat dissipation of the air layer behind the PV façade.

There is lots of research in this area. Since the 1990s, Yang et al. [10] have installed PV modules on the surface of buildings to establish PV walls and the numerical heat transfer models were established, and the impact of PV modules on indoor heating and air conditioning loads were studied. Ji et al. [11] undertook further research on this basis. The results show that when there is a gap between the PV module and the wall for natural ventilation, the heat dissipation of the PV module has an obvious effect, and the power generation efficiency is greatly improved. In the study by M. Fossa et al. [12] and Sarra et al. [13], it further shows that it has a good effect on reducing the heat gain of the room and improving the energy efficiency of the PV system. In order to give the PV double-skin façades (PV-DSF) better heat dissipation, many scholars have studied the size of the middle air layer (air channel or cavity, air gap/channel) in an attempt to find a key aspect ratio (air layer depth/air layer length) [14]. A. Zollner et al. [15] conducted an experimental study on the PV-DSF with air layer thicknesses of 0.3 m, 0.6 m and 0.9 m, and analyzed the influence of solar radiation on the turbulence intensity in the air layer. The results show that the influence depends on the ratio of the height of the air layer to the width, and the value of the relevant dimension design is given in the form of a relational function between the Nusselt number and the Archimedes number. Peng et al. [16] established a one-dimensional unsteady heat transfer model to evaluate the thermal performance of the multi-layer PV façade and found that the optimal thickness of the air layer in a PV-DSF is 0.06 m. In addition, the author also established a PV-DSF experimental platform with an air layer thickness of 0.4 m, the solar heat gain coefficient and the temperature at each position of the wall under different weather conditions and various operating modes (ventilated or non-ventilated) are obtained. Compared with the double skin façades with low-e layer, even the non-ventilated PV-DSF can reduce the solar heat gain by about 40%, which makes PV-DSF more suitable in subtropical climates [17]. A.S. Kaiser et al. [18] conducted an experimental study on the mechanical ventilation of the air layer in the PV curtain wall. It was found that when the aspect ratio of the air layer was 0.11, the component overheating (the difference between the actual heat dissipation and the theoretical heat dissipation) was minimized, when the mechanical ventilation speed is 6 m/s, the efficiency of the PV module can be increased by 19% compared with natural ventilation (0.5 m/s). Several studies on air flow and heat transfer in PV-DSF system have been carried out using computational fluid dynamics (CFD) [19–22].

However, the above studies did not discuss the length of the air layer in the vertical direction. According to the chimney effect theory, one way to enhance the natural convective heat transfer effect in the air layer is to increase the height difference of the inlet and outlet of the air layer, but this will also increase the temperature difference in the vertical direction of the PV-DSF, causing the operating temperature of the modules in the PV array to be non-uniform, thereby affecting the energy efficiency of the PV-DSF system. Therefore, it is necessary to consider the uniformity of the temperature field distribution

of the PV-DSF array in the vertical direction. Cristina et al. [23] and Sánchez et al. [24] proposed an open-joint ventilated façade (OJVF), in which the temperature distribution between each slab surface or the vertical slab is more uniform than the ordinary PV façade. This is because the OJVF has more ventilation openings than the ordinary PV facade, so it has better heat dissipation effect.

At present, the research combining PV modules with OJVF and DSF has not been seen. Therefore, a novel structure is proposed, in which PV modules are used as exterior sheet for open-joint PV ventilated double skin façades (OJ-PV-DSF). For OJ-PV-DSF, the temperature of the exterior surface (PV module) is an important parameter, which not only directly affects the power generation of the PV array, but also indirectly affects the energy characteristics of the integrated PV wall system. As we know, in the OJ-PV-DSF system, the opening of a larger ventilation joint will cause better heat dissipation for PV modules, but it will reduce the available area that can be used to install PVs. Therefore, there should be a suitable opening size to take into account both heat dissipation and the available area.

## 2. Description of Methodology

### 2.1. Basic Theory and Equation

Compared with PV-DSF, OJ-PV-DSF has a more open joint, which makes its air flow and heat transfer process more complicated. In the OJ-PV-DSF system, the heat transfer process in the cavity involves solar radiation, heat conduction, radiation heat transfer and a natural convection heat transfer process, but there is a large difference in the flow behavior within the cavity. Due to the presence of more open joint, air can freely enter and exit the cavity, causing the air flow at the inlet and outlet of open joint to entangle, especially when the PV module is used as an exterior sheet, the thermal changes caused by such complex convection will strongly affect the electrical performance of the PV module. For this flow and heat transfer phenomenon, a two-dimensional or even three-dimensional differential equation should be used for detailed study. In this paper, two-dimensional governing equations are used to model the flow and heat transfer of OJ-PV-DSF system.

### 2.2. Description of Computational Fluid Dynamics (CFD) Modeling

In this paper, the study of air flow and heat transfer in OJ-PV-DSF system is mainly carried out by CFD. Considering that there are many opening joints in the OJ-PV-DSF system, and it is not clear before the modeling that the opening joints are air inlets or outlets, it is necessary to expand the computational domain, which is consisting of the building, PV module and the outdoor atmosphere, and therefore more accurately reflect the entire flow field around the OJ-PV-DSF. By simulating the flow state of the outdoor atmospheric boundary layer, the flow field around the PV module and the middle air layer under the combined action of wind pressure and hot pressure is realized. The determination of the computational domain is according to the guidelines of [25], as shown in Figure 1.

In this model, considering the large differences in geometric dimensions between PV modules, walls, and outdoor atmospheres, partitioning is required for meshing. The grid size of the outdoor atmosphere is controlled at 0.05–0.02 m, the wall is controlled at 0.01–0.005 m, and the PV module and aluminum edging are controlled at 0.5–2.0 mm. In addition, the outer surface of the PV module and the outer surface of the wall need to be studied intensively. Therefore, structured grids are used, and grid refinement is applied to further improve the calculation accuracy, as shown in Figure 2. The grid quantity is verified via a further increase in the number of nodes with no effect on the temperature in the simulation results. In the experimental model, there are about 5 million; in case study model, there are about 23 million.

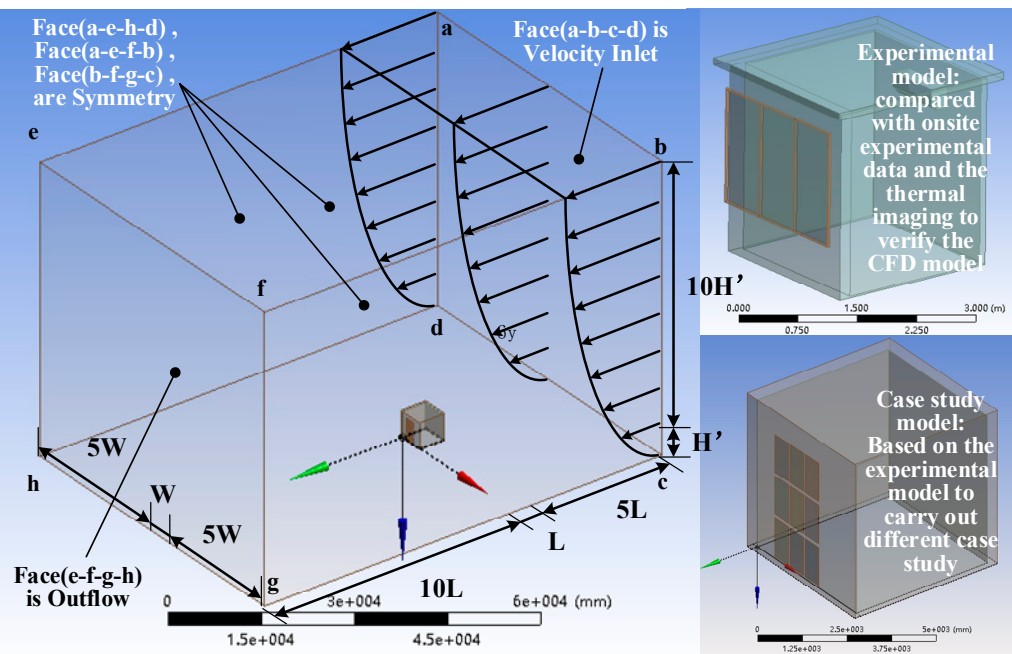

**Figure 1.** The computational domain and the open-joint photovoltaic double-skin facade (OJ-PV-DSF) model.

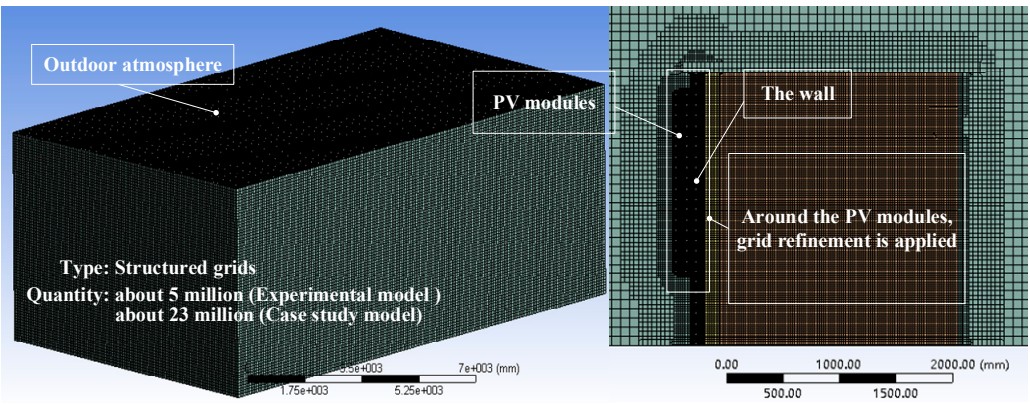

**Figure 2.** Mesh of the modeling.

In the OJ-PV-DSF system, since the flow of air at the open nodes in the air cavity is complex, swirling or curved wall flow may occur, so the RNG (renormalization-group) $k - \varepsilon$ model [20,21] is used. For the research on the radiation heat transfer in the OJ-PV-DSF system, the key point is to consider the radiation heat transfer between the surfaces, the DO (discrete ordinates) model is selected for solving the radiation problem in this study.

### 2.3. Experimental Model

In order to verify the accuracy of the CFD model, a comparative experimental study was carried out. As shown in Figure 3, the simulation model is established according to the real size of the PV modules and the various parts that mounted on the west side of the experimental room. The thickness of the air layer between the PV module and the building wall is 150 mm, which is equal to 0.1 of the vertical length of the PV module, and the air layer is connected to the atmosphere without any blockage and the dimensions of other parts are as shown in Figure 3a. In order to more accurately reflect the temperature at the edge of the PV module, the aluminum cladding part of the PV module was carefully

modeled. The existence of the air chamber in the middle of the aluminum profile was fully considered and shown in Figure 3c.

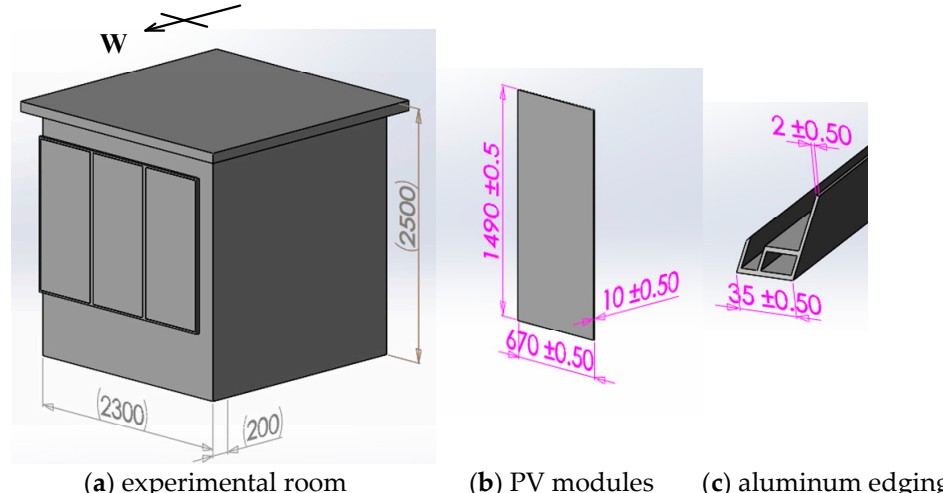

(**a**) experimental room          (**b**) PV modules          (**c**) aluminum edging

**Figure 3.** The model of the experiment room, PV module and aluminum edging.

The steady state calculation is used in the simulation, and the corresponding moment is around 14:30 on 2 July, so the corresponding meteorological parameters at the moment are input for simulation as shown in Table 1. These meteorological parameters were tested by the experimental system. Among them, the measuring point of wind speed is at the height of 3.5 m above the ground of the experimental cabin recorded by the meteorological data collection system is established on site.

**Table 1.** The input meteorological parameters during simulation.

| Wind Velocity | Wind Direction | Outdoor Temperature | Atmospheric Pressure | Total Radiation | Diffuse Radiation |
|---|---|---|---|---|---|
| 2.0 m/s | 170 | 33.1 °C | 100.3 kPa | 715.4 W/m$^2$ | 172.3 W/m$^2$ |

*2.4. Analysis of Experimental Results*

2.4.1. Comparison with Measured Temperature

In order to verify the accuracy of the simulation results, at the same time as the simulation, the temperature values of the backplane of the PV module, the air layer and the outer surface of the wall in the experimental room at three different heights were measured (the parameters of temperature sensors are shown in Table 2) and compared with the simulated values. The corresponding results are shown in Figure 4 and Table 3. It can be seen that the simulation results of the temperature of the PV backplane are relatively accurate, but the errors of the simulation results for the temperature of the outer surface of the wall and the air layer are relatively large, the mean absolute error being about 1.5 K.

**Table 2.** Parameters of temperature sensors and the thermal imaging equipment in the experimental study.

| Name | Type | Accuracy | Temperature Range | Others |
|---|---|---|---|---|
| Thermal resistance | Pt100 | A class (±0.15 + 0.002 °C) | −50–200 °C | - |
| Thermal imaging equipment | NEC R300SR | ±1.0 °C | −40–500 °C | Temperature resolution: 0.03 °C |

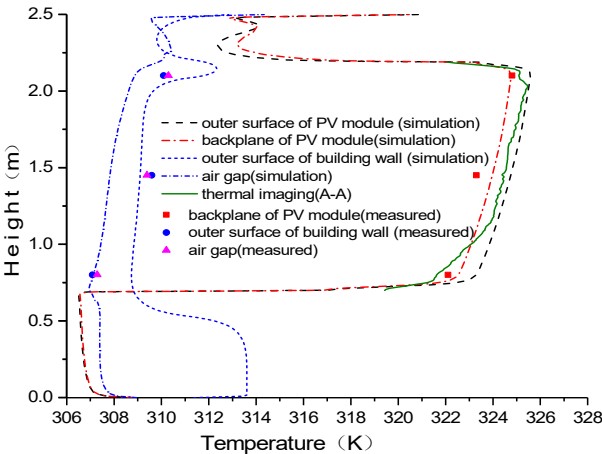

**Figure 4.** The simulation results are compared with the measured results.

**Table 3.** The comparison between the simulated temperature and the measured temperature.

| Height(m) | Project | Back Surface of PV Module | Outer Surface of Wall | Air Gap |
|---|---|---|---|---|
| 0.80 | measured (K) | 322.10 | 305.10 | 308.50 |
| | simulated (K) | 322.53 | 308.73 | 307.12 |
| 1.45 | measured (K) | 323.30 | 309.30 | 309.40 |
| | simulated (K) | 323.97 | 309.10 | 307.83 |
| 2.10 | measured (K) | 323.90 | 310.10 | 310.30 |
| | simulated (K) | 324.76 | 311.71 | 308.87 |
| Mean absolute error (K) | | 0.65 | 1.81 | 1.46 |

### 2.4.2. Comparison of the Temperature Field with Thermography

With thermal imaging equipment (the parameters are shown in Table 2), the temperature distribution of the surface of the PV module in the experimental room was collected. The distance from the camera to OJ-PV-DSF was 2.5 m, and the emissivity factor of the PV modules was 0.85 via laboratory test. The corresponding results are shown in Figure 5a, and the temperature distribution of the PV module calculated by CFD is shown in Figure 5b. For the overall distribution of the temperature field, qualitatively speaking, the temperature distribution calculated by CFD and the actual thermal imaging results are approximately the same for the middle and the right in the three PV modules. For the PV module on the left side, the temperature at the left edge in particular is somewhat different from the actual thermal imaging temperature. This difference may come from local air flows around the left edge that differ from the simulation results and the simulation does not show that the local wind direction is blowing from right to left in the image. However, in this study, the overall temperature distribution of the PV arrays is considered more important, so subsequent analysis is based on the temperature on the center line of the module in the middle. Therefore, this error was acceptable for the study on the influence of the arrays' MD on cell temperature in the OJ-PV-DSF system.

In addition, quantitative analysis is also undertaken by extracting the temperature value of the middle PV module in the thermal imaging picture, as shown in Figure 5a, then compares it with the temperature at the same location as the simulation result, as shown in Figure 4, the simulation temperature is slightly higher than the temperature value of thermal imaging obtained of about 0.2–0.3 K, for the PV module located in the middle can be considered as the temperature simulation results and the actual situation close to each other.

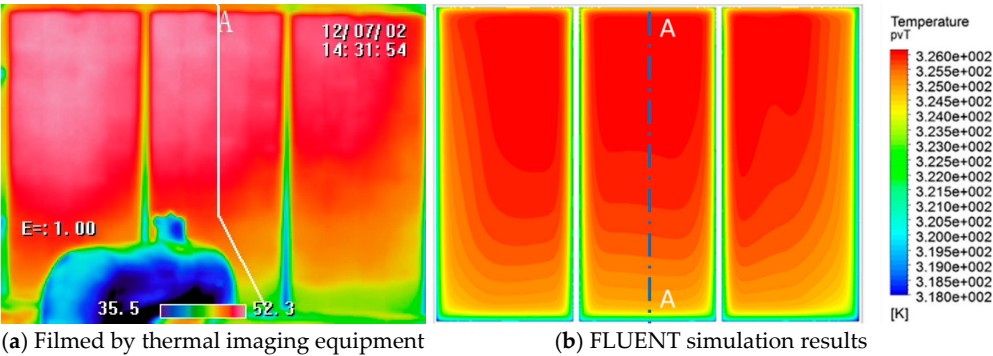

(**a**) Filmed by thermal imaging equipment          (**b**) FLUENT simulation results

**Figure 5.** Comparison between the simulation results with the thermal imaging results.

Therefore, combining the above comparison with the measured data and the thermal imaging temperature distribution, the CFD model is accurate and can be used for the next analysis.

## 3. Simulation of Modules Temperature at Different Mounting Dimensions

### 3.1. Introduction to the Mounting Dimensions (MD) of the Open-Joint

In the OJ-PV-DSF system, the opening joint is mainly present between the adjacent two PV modules, that is, the spacing between two PV modules adjacent in the horizontal direction and the vertical direction. In addition, the spacing between the PV modules and the wall will also greatly affect the temperature of the PV modules and even the performance of the OJ-PV-DSF system. Therefore, only the above three MD of the OJ-PV-DSF system are considered in this study. In order to more concisely study the influence of these three MD on the temperature of the PV modules, three dimension-factor are defined in this study. The distance between PV modules and the wall: $a = D/H$ (see Appendix A, Table A2 ) is used to denote the size of this distance relative to the length of the PV module in the vertical direction, the distance between two adjacent PV modules in the vertical direction: $b = H_g/H$ is used to denote the magnitude of this distance relative to the PV module's length in the vertical direction, the distance between two adjacent PV modules in the horizontal direction: $c = Lg/L$ is used to denote the magnitude of this distance relative to the width of the PV module in the horizontal direction. As shown in Figure 6, which is a schematic diagram of the relevant distance mentioned above.

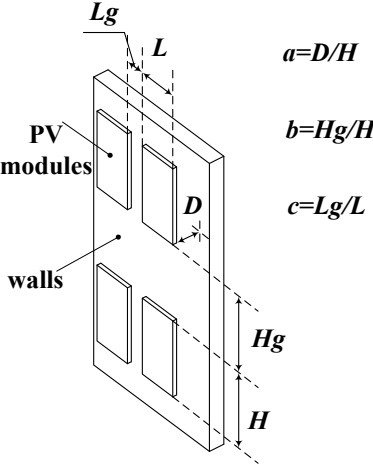

**Figure 6.** The schematic diagram of the *a*, *b* and *c*.

### 3.2. The Distance between Photovoltaic (PV) Modules and Wall

In order to study the temperature variation of PV wall under different pitches, a 3 × 4 PV array as shown in Figure 7 is established. In the figure, keep *b* = 0.1 and

$c = 0.1$, and change *a* from 0.05 to 0.3, the corresponding PV module surface temperature changes. Considering that, in a PV system, the pick temperature of an individual PV module and the uniformity of the array temperature will affect the overall efficiency [5,6], in this study, the maximum temperature and the temperature distribution of the PV modules in the PV array are worked as performance measures. From Figure 7a–f, it is obvious that the temperature of the PV module in the whole case is significantly higher than that of the other five cases when *a* = 0.05, and the temperature in the vertical direction also increases obviously faster than other examples. As the value of *a* increase, the overall temperature of each case gradually decreases, and the temperature difference in the vertical direction also decreases obviously. When *a* = 0.30, the temperature difference in the vertical direction is not obvious.

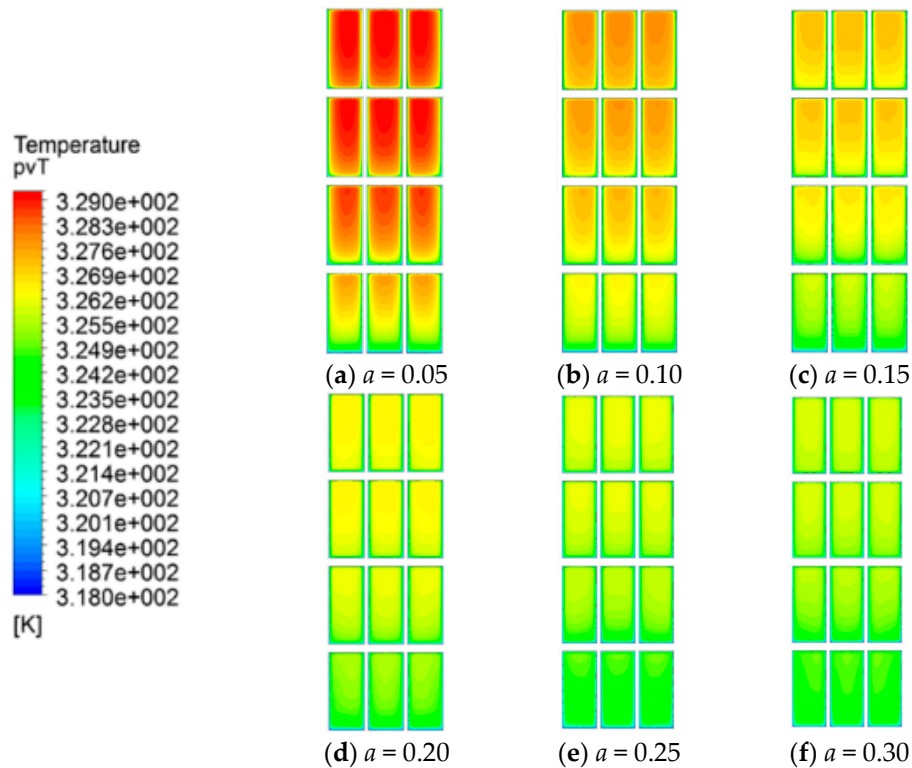

**Figure 7.** The temperature field of the photovoltaic (PV) array under different *a* values.

As shown in Figure 8, when *a* is a different value, the temperature on the axis of outer surface of each PV array varies with height. Under different cases, the temperature of the PV modules in the array increases as they rise in height. When *a* = 0.05, the maximum temperature of the uppermost PV module can reach 329.2 K, and the temperature difference of the outer surface of the array is about 5 K; while when *a* = 0.10, the maximum temperature of the PV array is 327.9 K, and the temperature difference is reduced to about 3.5 K. Furthermore, when *a* = 0.15, the maximum temperature and the difference are reduced to 327.2 K and 2.5 K, respectively. Finally, when *a* = 0.30, the maximum temperature of the module is 326.3 K, and the overall temperature distribution is more uniform, the temperature difference between the top and bottom of the module is only about 2 K. It can be seen that the maximum temperature drop (1.3 K) is most obvious when *a* is change from 0.05 to 0.10, about a reduction of 45% compared to the value (2.9 K) when there is a changes from 0.05 to 0.30. When *a* is changed from 0.05 to 0.15, about a reduction (2.0 K) of 69% in the whole reduction from 0.05 to 0.30, then the drop is gradually reduced with increasing *a*.

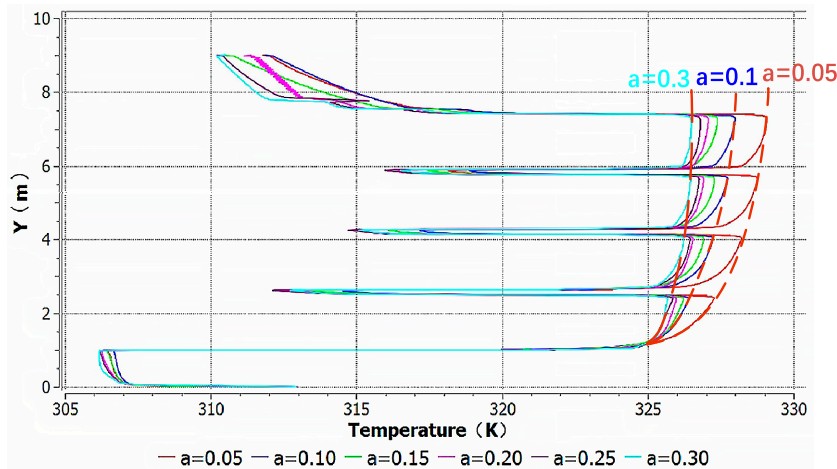

**Figure 8.** The temperature of the outer surface of PV module under different *a* values.

As shown in Figure 9, when *a* is a different value, the temperature on the axis of the outer surface of the wall covered by PV modules in each case changes with height. When *a* changes from 0.05 to 0.30, the maximum temperature of the wall under the PV arrays reduced most significantly for the 1.1 K, but when *a* changed from 0.05 to 0.1, the wall temperature reduced the most greatly for the 0.7 K, which was relatively obvious. The temperature difference generated in the vertical direction is reduced from 0.7 K (when a = 0.5) to 0.4 K.

Figure 10 shows the change of air layer temperature under different *a* values. It can be seen that the change is obvious. When *a* = 0.05, the air layer is too narrow, the influence of air entering the opening is obvious and its temperature fluctuates, and as *a* increase, the influence decreases gradually. In addition, as *a* increases, the temperature drop of air layer decreases gradually. When *a* changes from 0.05 to 0.10, the temperature in the upper part of the air layer decreases most obviously, reaching 2.4 K, and then decreasing by about 2.0 K when *a* changes from 0.10 to 0.15. At this time, the temperature drop has dropped to about 65% of that at 0.30. In summary, when the value of *a* is 0.15, it has the most obvious effect on the temperature drop of the PV module and the air layer. However, with the value of *a* increasing further, this influence is minimal, but it reduces the available area of the building, thus reducing the overall effectiveness of the OJ-PV-DSF system.

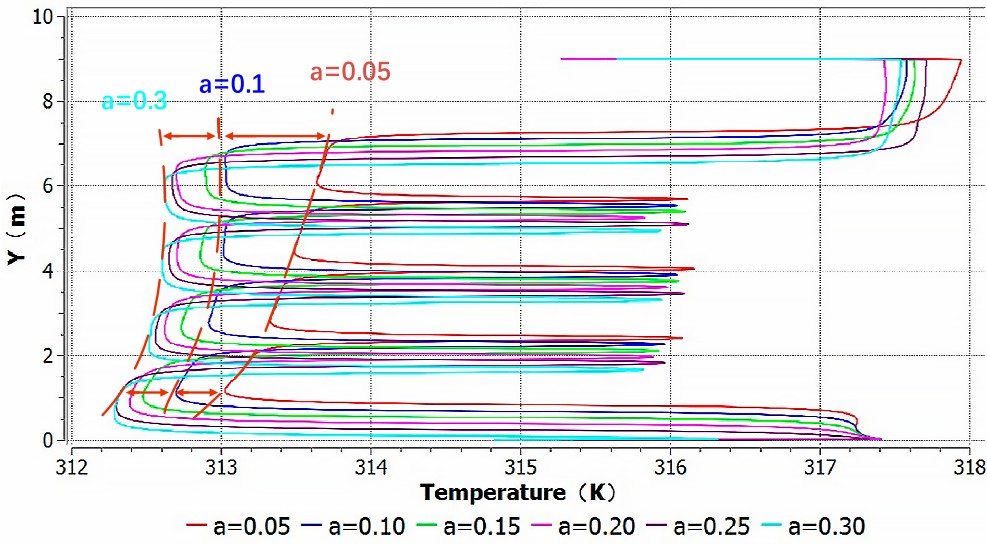

**Figure 9.** The temperature of the outer surface of wall under different *a* values.

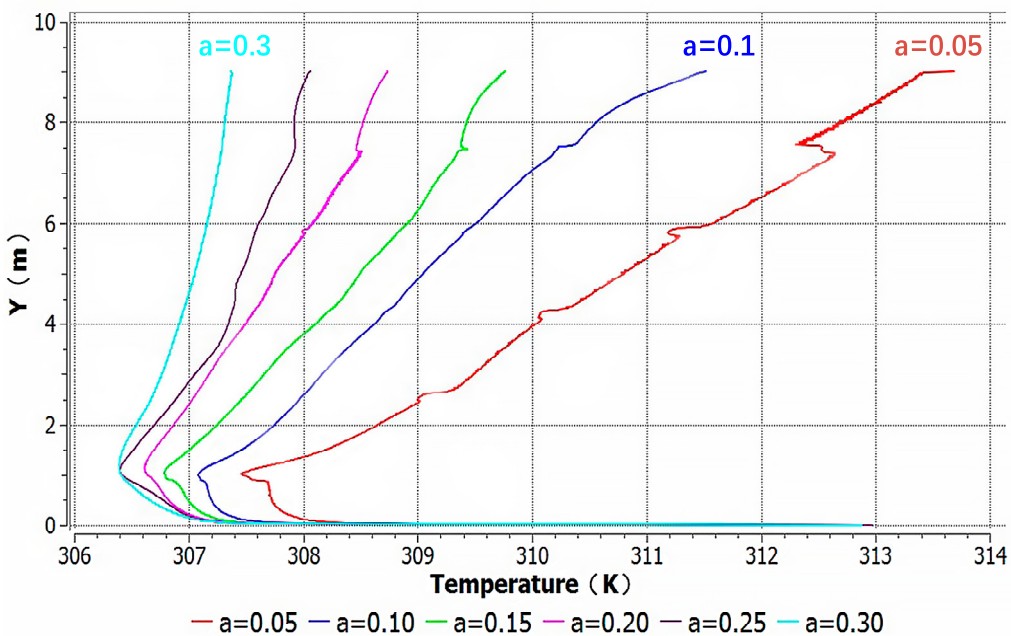

**Figure 10.** The temperature of the air layer under different *a* values.

*3.3. The Distance between Two Adjacent PV Modules in the Vertical Direction*

As shown in Figure 11, if we keep *a* = 0.1 and *c* = 0.1, and change *b* from 0 to 0.3, the corresponding PV module surface temperature changes. From Figure 11a–d, it is obvious that the temperature of the PV modules in the whole case is higher than that of the other several cases when *b* = 0, and the temperature in the vertical direction increases also obviously faster than in other cases. With the increase of *b* value, the overall temperature of each case decreases gradually, the temperature difference in the vertical direction also decreases obviously, and the temperature drop is the most obvious when *b* changes from 0 to 0.10. It can reach 1.1 K. then the temperature difference in the vertical direction is not obvious with *b* changing from 0.10 to 0.30. Therefore, the *b* = 0.10 may be preferable to reducing the surface temperature of the PV modules.

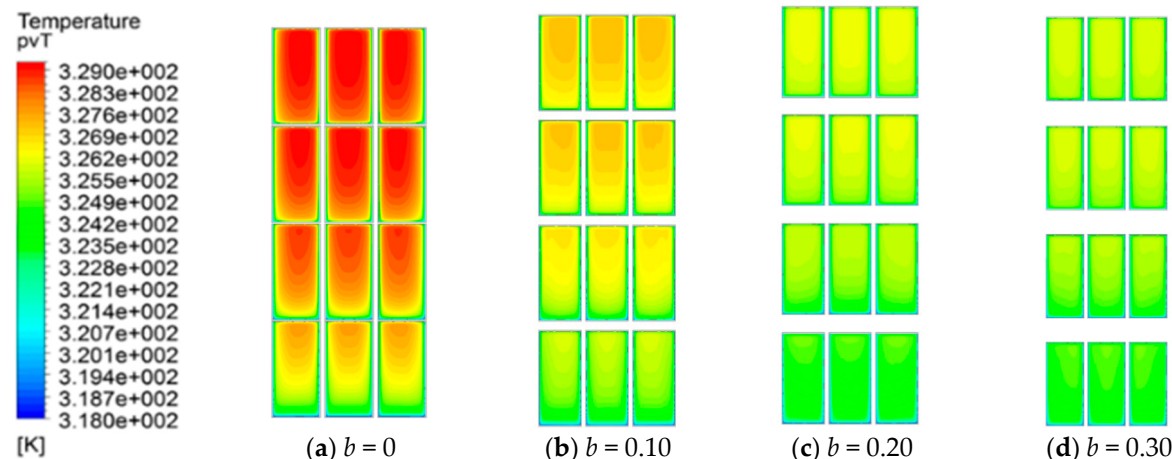

**Figure 11.** The temperature field of the PV array under different *b* values.

*3.4. The Distance between Two Adjacent PV Modules in the Horizontal Direction*

As shown in Figure 12, keep *a* = 0.1 and *b* = 0.1, and change *c* from 0 to 0.3, the corresponding PV module surface temperature changes. From Figure 12a–d, it can be seen that when *c* = 0, the temperature of the PV module in the whole case is obviously higher

than the other several cases, and the temperature of the PV module in the middle of the horizontal direction is the highest. As *c* changes from 0 to 0.10, the overall temperature of each case decreases gradually and the temperature difference in the vertical direction also decreases significantly. However, as the *c* increases further, the temperature change of the entire PV array is not obvious. This conclusion can also be drawn from Figure 13, when *c* changes from 0 to 0.1, the temperature on the central axis of the outer surface of the PV module decreases significantly, while the temperature does not substantially change when *c* is further increased. Therefore, leaving a gap in the horizontal direction will help the overall cooling of the PV wall. However, when the value of *c* is greater than 0.1, it will reduce the effective area of the PV array.

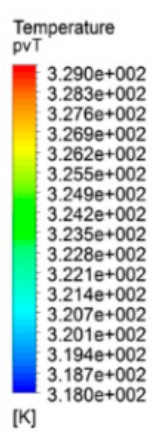
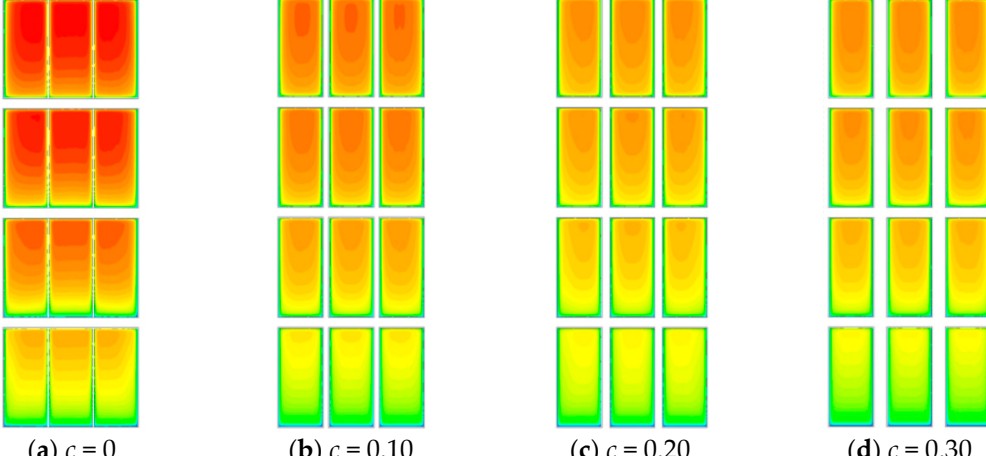

**(a)** *c* = 0      **(b)** *c* = 0.10      **(c)** *c* = 0.20      **(d)** *c* = 0.30

**Figure 12.** The temperature field of the PV array under different *c* values.

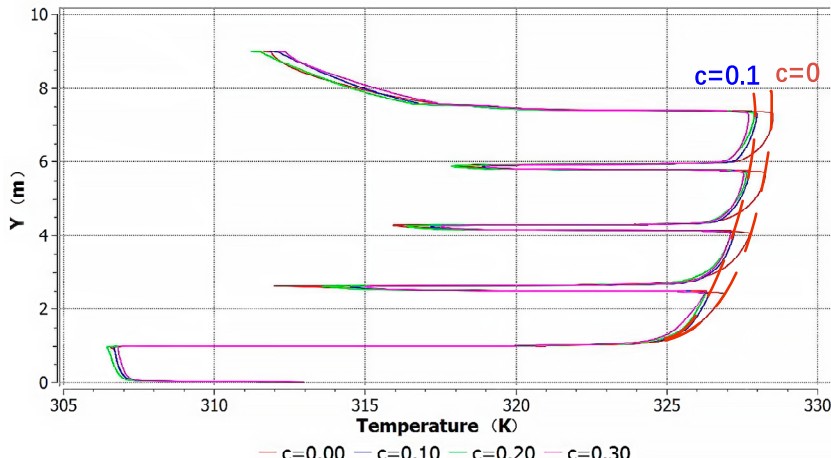

**Figure 13.** The temperature of the outer surface of PV module under different *c* values.

## 4. Conclusions

In this paper, a three-dimensional CFD model was established to study the influence of MD of a PV array on module temperature in OJ-PV-DSF. The results show that:

(1) The MDs of PV do have a large influence on the temperature of the PV modules. When the ratio between the distance between the module and wall and the length of the module in the vertical direction changes from 0.05 to 0.15, the temperature drop of the PV module is the most obvious, reaching 2.0 K, and with the value of *a* is further increased, the temperature change of the PV modules is small.

(2) When the ratio of the distance between two adjacent PV modules in the vertical direction and the length of the module in the vertical direction (defined as *b* value)

is changed from 0 to 0.1, the temperature drop of the PV module is most obvious, reaching 1.1 K. When the ratio of the distance between two adjacent PV modules in the horizontal direction and the length of the module in the horizontal direction (defined as *c* value) is changed from 0 to 0.1, the temperature of the PV module is lowered by 0.8 K, and when *c* is further increased, the temperature change of the PV array is not significant.

(3)   It is recommended to maintain the three-dimension factor *a*, *b*, and *c* of the PV array at 0.1–0.15, 0.1, and 0.1, respectively, thereby improving the energy efficiency of integrated PV buildings.

**Author Contributions:** Data curation, T.G.; Formal analysis, S.M.; Project administration, J.Z.; Software, T.G. and Y.Z.; Writing—original draft, W.Z.; Writing—review & editing, W.Z. All authors have read and agreed to the published version of the manuscript.

**Funding:** This research was funded by the National Natural Science Foundation of China, Grant No. 51908287 and the National Natural Science Foundation of Jiangsu Province, Grant No. BK20180484.

**Institutional Review Board Statement:** Not applicable.

**Informed Consent Statement:** Not applicable.

**Data Availability Statement:** Not applicable.

**Acknowledgments:** This study was supported by the National Natural Science Foundation of China (Grant No. 51908287) and the National Natural Science Foundation of Jiangsu Province (Grant No. BK20180484).

**Conflicts of Interest:** The authors declare no conflict of interest.

## Appendix A

**Table A1.** A list of abbreviations.

| Abbreviation | Meaning |
| --- | --- |
| MD | Mounting dimensions (in this paper, it means the distance between PV modules and walls, and the distance between tow modules adjacent in the vertical or horizontal direction in PV arrays on the surface of the building façade) |
| PV | Photovoltaic |
| PV-DSF | PV double skin façades |
| CFD | Computational fluid dynamics |
| OJVF | Open-joint ventilated façade |
| OJ-PV-DSF | Open-joint PV ventilated double skin façades |
| DO | Discrete Ordinates |
| RNG | Renormalization-group |

**Table A2.** A list of symbols.

| Symbols | Meaning |
| --- | --- |
| *a* | the ratio of the distance between PV modules and the wall to the vertical length of PV modules |
| *b* | the ratio of the distance between two adjacent PV modules in the vertical direction and the length of PV modules in the vertical direction |
| *c* | the ratio of the distance between two adjacent PV modules in the horizontal direction and the width of PV modules in the horizontal direction |
| *D* | the distance between PV modules and the wall |
| *Hg* | the distance between two adjacent PV modules in the vertical direction |
| *H* | the Length of PV module in vertical direction |
| *H'* | the height of the wall |
| *W* | the width of the wall |
| *Lg* | the distance between two adjacent PV modules in the horizontal direction |
| *L* | the width of PV modules in the horizontal direction |

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
