# Peer review of "Study on the Influence of Mounting Dimensions of PV Array on Module Temperature in Open-Joint Photovoltaic Ventilated Double-Skin Façades"

_sustainability, doi:10.3390/su13095027_

Round 1
Reviewer 1 Report
Comments on “Study on the Influence of Mounting Dimensions of PV Array on Modules Temperature in Open-Joint Photovoltaic Ventilated Double Skin Facades”
Dear Authors,
The paper must be significantly improved. Please consider the following remarks:
Major comments:
(1) The Abstract should be re-written. Please add the main condition of research.
(2) Keywords. Please improve - Authors should make better use of this section to allow the article to be found on search engines.
(3) I can not find any new scientific method. The novelty of the paper ought to be indicated in the context. The background also should be extended with broader literature studies exposing the theoretical and practical need of the research. Please compare your approach to another from references.
(4) CFD part: The major lack of the performed research is the lack of proper validation.
For the CFD should be presented: time step for the numerical calculation (and its influence on obtained results) and the influence of other turbulence models on the obtained results (in comparison to chosen one).
Please add the meshing procedure, type of selected mesh. Please explain the reason why you selected the number of mesh elements. Please define: boundary conditions, simulated method for boundaries. Please add analyses of the influence of the meshing procedure on the obtained results (in comparison to chosen one).
Did you do multi-parameters model in Ansys Fluent? Please add table with range of parameters.
(5) Please compare results with other from reference.
(6) Figure 2 should be significantly enlarged with an aim that meshing details can be clearly seen.
Minor comments (answers are not necessary):
(1) Line 27. Please improve the way of calling reference.
(2) Line 71. From the SI Brochure, §5.3. 3: "The numerical value always precedes the unit, and a space is always used to separate the unit from the number." The same approach I suggest applying also in the rest of the manuscript.
(3) Please explain all nomenclature and abbreviations in one place.
(4) Line 183-184. Please explain “the building wall is 150 mm”
(5) Table 1. Please add source
(6) Figure 5. Please explain the difference (temperature) between (a) and (b) part of figure.
(7) Please improve the reference part. Please use Sustainability journal template.
(8) Figure 7. One colorscale area of legend is enough. Please make larger all figures.
Some typos.
Table 1. please change “KPa” -> “kPa”
Line 262 – 263
Line 310 “1.1”
Author Response
(1) The Abstract is re-written.
In building integrated photovoltaics, it is important to solve the heat dissipation problem of photovoltaic modules. In this paper, the CFD method is used to simulate the flow field around the Open-Joint Photovoltaic Ventilated Double Skin Facades (OJ-PV-DSF) to study the influence of the mounting dimensions of PV array on the module temperature. The typical summer after-noon meteorological parameters, such as the total radiation (715.4 W/m2), the outdoor temperature (33.1 ℃), and the wind speed (2.0 m/s), etc., are taken as input parameters, with the DO (Discrete Ordinates) model and the RNG(Renormalization-group) model, a steady state calculation is carried out to simulate the flow of air in and around the cavity under the coupling of hot pressure and wind pressure, thereby obtained the temperature field of the PV array and the wall. In addition, the simulation results are compared with the onsite experiment data and thermal imaging to verify the accuracy of the CFD model. Then three mounting dimensions of open joints are discussed in this study. The results show that when the a value (represents the distance between PV modules and wall) changes from 0.05 to 0.15, the temperature drop of the PV module is the most obvious, reaching 2.0 K. When the b value (represents the distance be-tween two adjacent PV modules in the vertical direction) changes from 0 to 0.1, the temperature drop of the PV module is most obvious, reaching 1 K. When the c value (represents the distance between two adjacent PV modules in the horizontal direction) changes from 0 to 0.1, the temperature of the PV module is lowered by 0.8 K. Thus, a = 0.1~0.15, b = 0.1 and c = 0.1 are recommended for engineering applications to effectively reduce the module temperature.
(2)The keywords are re-written.
Photovoltaic double skin facades (PV-DSF); Air layer; PV array arrangement; CFD; Module temperature
(3)In PV-DSF, it is important to solve the heat dissipation problem of photovoltaic modules.
In this study, PV modules are used as exterior sheet for open-joint PV ventilated double skin facades (OJ-PV-DSF) to improve the effect of heat dissipation. Similar studies have not been seen.
And compared with PV-DSF, OJ-PV-DSF has more open-joint, which makes its air flow and heat transfer process more complicated. In the OJ-PV-DSF system, the heat transfer process in the cavity is both involve solar radiation, heat conduction, radiation heat transfers and natural convection heat transfer process, but there is a large difference in the flow behavior within the cavity. Due to the presence of more open joint, air can freely enter and exit the cavity, causing the air flow at the inlet and outlet of open joint to entangle, especially when the PV module is used as an exterior sheet, the thermal changes caused by such complex convection will strongly affect the electrical performance of the photovoltaic module. By simulating the flow state of the outdoor atmospheric boundary layer, the flow field around the PV module and the middle air layer under the combined action of wind pressure and hot pressure is realized.
The CFD method is used to simulate the flow of air in the cavity under the coupling of hot pressure and wind pressure, thereby obtained the temperature field of the PV array and the wall.
The introduction is re-written. And the background is extended with broader literature studies exposing the theoretical and practical need of the research.
(4) CFD part:
a)The steady state calculation is used in this study. So we didn’t consider about time step for the numerical calculation.
- b) The onsite experimental study and thermal imaging equipment were carried out to verify the accuracy of the CFD model.
- c) As for the selection of turbulence models, relevant literatures were referred.
- d) The meshing procedure are added. The meshing quantity is verified via further increase the number of nodes with no effect on the temperature in the simulation results. In experimental model, there are about 5 million; in case study model, there are about 23 million.
- e) The boundary conditions, simulated method for boundaries are shown as reedited Fig 1
Figure 1. The computational domain and the OJ-PV-DSF model.
- f) In this study, we didn’t use multi-parameters model. Different case study was carried out by manually modifying the relevant geometric dimensions.
(5) There are no comparable studies for OJ-PV-DSF. So it is hard to compare results with other from reference.
(6) Fig 2 is reedited as shown as follow:
Figure 2. Mesh of the modeling.
“Minor comments” are all modified.

Reviewer 2 Report
Review
Manuscript ID: sustainability-1146472
Article: Study on the Influence of Mounting Dimensions of PV Array 2 on Modules Temperature in Open-Joint Photovoltaic Venti-3 lated Double Skin Facades
The article presents the analyses of mounting dimensions of the photovoltaic array in the open-joint photovoltaic ventilated double skin facades (OJ-PV-DSF) to reduce the array temperature and thus increase their energy efficiency. The authors developed a CFD model consisting of a building, a PV module and an outdoor atmosphere and correctly applied different mesh sizes depending on the model element being described. To validate the model, the temperature values of the PV module backplate, air layer and external wall surface in the experimental room were measured at three different heights and compared with the simulation results. In addition, thermographic measurements were used. However section 2.4 does not specify the equipment and its measurement accuracy (used for temperature and thermographic measurements). The work requires proofreading in terms of language. In particular section 3.4 - which describes the simulation results.
In this regard, it is recommended that the article should be published in Sustainability Journal after major revision. The authors should consider the following points in order to improve the quality of the paper and before a final approval:
- Please complete the missing information on the measuring equipment used for temperature and thermographic measurements - section 2.4.( the type, accuracy of measurement)
- The weakest element of the publication is the description of thermal imaging measurements. A thermal imaging measurement is a function of five parameters: what values were used to calibrate the device (value of the emissivity factor, the distance from the camera to OJ-PV-DSF..)?
- Line 209 : Did the authors mean Mean Absolute Error (MAE) when they wrote " average absolute error”? What formula was used for the calculation?
- Figure 3: Placing the legend inside the graph makes it unreadable. Please change the legend`s location.
- Lines 262 and 263 - wrong order of description.
- Please improve the resolution of figure 8,9,10,13.
- Line 287-296 - Figure 8 description - not clear. “ When a is change from 0.05 to 0.15, about a reduction of 70% in the whole reduction, a drop of about 0.7K, then the drop is gradually reduced with increasing a”. Please rewrite the description of the whole point 3.2 The distance between PV modules and wall and 3.3 The distance between two adjacent PV modules in the vertical direction (line 310)
- Line 310 - incorrect numbering of sub-item (1.1) should be 3.3
- Please improve the description of the simulation results in terms of language. Inappropriate use of the word obvious (“With the increase of b value, the overall temperature of each case decreases gradually, and the temperature difference in the vertical direction also decreases obviously, and the temperature drop is the most obvious when change b from 0 to 0.10, then the temperature difference in the vertical direction is not obvious with change b from 0.10 to 0.30. Therefore, the best value of b may be about 0.10, the effect of reducing the surface temperature of the PV module is relatively more obvious”.)
- Why were 6 distances for parameter a analysed, and only 4 distances for parameters b and c?
- Line 322-323 on which basis it was concluded that the best value of b may be around 0.10?
- What does it mean that “the effect of reducing the surface temperature of the PV module is relatively more obvious”.

Author Response
- Table 2 has been added:
Table 2 Parameters of temperature sensors and the thermal imaging equipment in the experiment study
|
Name of the devices |
Type |
Accuracy |
Temperature range |
Others |
|
Thermal resistance |
Pt100 |
A class (±0.15+0.002℃) |
-50-200℃ |
- |
|
Thermal imaging equipment |
NEC R300SR |
±1.0℃ |
-40-500℃ |
Temperature resolution:0.03℃ |
- When processing the thermal image, the surface emissivity of PV modules is considered in the software. The value of the emissivity factor is 0.85 via onsite measured. And the distance from the camera to OJ-PV-DSF is 2.5 m.
The purpose of testing the temperature on the back side of the photovoltaic module is to assist in verifying the temperature of the thermal imaging.
- Yes. We mean “Mean Absolute Error (MAE)”. It has been corrected.
- May be fig5? It has been corrected.
- It has been corrected.
- The resolution of figure 8,9,10,13 are improved. As shown in the revision
- The whole section is rewritten, as shown in the revision.
- It has been corrected.
- The whole section is rewritten, as shown in the revision.
- a is used to denote the size of this distance relative to the length of the PV module in the ver-tical direction, which shows the distance between PV modules and the wall. And it varies from 0.5 to 3.0 with an interval of 0.5, so there are 6 distances.
b is used to denote the magnitude of this distance relative to the PV module's length in the vertical direction, which shows the distance between two adjacent PV modules in the vertical direction; and c is used to denote the magnitude of this distance relative to the width of the PV module in the horizontal direction, which shows the distance between two adjacent PV modules in the horizontal direction. b and c vary from 0 to 3.0 with an interval of 1.0, so there are only 4 distances, respectively.
- It's not clear in the original manuscript. And the whole section is rewritten, as shown in the revision.
- It's not clear in the original manuscript. And the whole section is rewritten, as shown in the revision.

Reviewer 3 Report
The authors presented the size effects of PV array modules through CFD simulated with a experimental calibration. In general, the manuscript is poorly written. There are many grammar errors and awkward sentences, starting from abstract over the entire manuscript. The authors have to revise the entire manuscript to make it more readable.
Specifically, the authors should improve the manuscript as follows:
1) Title: it is confusing using the word of "mounting dimensions of.." and over the entire manuscript, and please revise it.
2) abstract: aside from the grammar errors, the sentences from line 14-21 cannot provide clear information and the authors have to revise it to ensure what are your findings and contributions.
3) Introduction: There are many grammar errors and please carefully proofread them. Also try to avoid using subjective descriptions. The last paragraph supposes to provide the clear plan for this study, not back to literature review.
4) Section 2. It seems the basic theory is not derived from the authors and please provide citations for Eqn. 1-3. Also, the authors did not provide any connection of these equations to Section 2.2, and it raises the question: why would we need Section 2.1?
5) Section 3. The authors defined three factors, a, b and c, and discussed each of them by three sections. One question is what is the main goal, optimal performance? The following question is what is performance measures, not just based on contour color or peak value?
6) Conclusion: this section should be totally revised to ensure it summarized the clear contribution, not another discussions.
Author Response
1)“mounting dimensions” is explained in the new added “Table 4. A list of abbreviations.”
Mounting dimensions (in this paper, it means the distance between PV modules and walls, and the distance between tow modules adjacent in the vertical or horizontal direction in PV arrays on the surface of the building façade).
2) The abstract is re-written.
3) This part is re-written.
4) Equation 1~3 is the fundamental equation. There is no need to list it here. So this part is deleted in this revision.
5) In this study, the maximum temperature and the temperature distribution of the PV modules in PV array are worked as performance measures. Because, in a PV system, the pick temperature of an individual PV module and the uniformity of the array temperature will affect the overall efficiency. So the contour color and peak value of the temperature are important. The above explanation has been added to the revision, as shown in Line 261~263.
6) The conclusion is re-written.

Round 2
Reviewer 1 Report
Minor comments:
(1) Line 32. Please improve the way of calling reference.
(2) Please explain all nomenclature and abbreviations in one place. Please use alphabetical order. Please add for example, dimension-factor a, b, and c; Hg, H, H’, W, L
(3) Please improve the reference part. Please use Sustainability journal template.
Author Response
“Minor comments” are all modified.
Reviewer 2 Report
The relevant changes have been made.
Author Response
Thank you very much for your comments and suggestions.